# Prevalence of Chronic Fatigue Syndrome (CFS) in Korea and Japan: A Meta-Analysis

**DOI:** 10.3390/jcm10153204

**Published:** 2021-07-21

**Authors:** Eun-Jin Lim, Chang-Gue Son

**Affiliations:** 1Department of Integrative Medicine, Graduate School of Integrative Medicine, CHA University, 120 Haeryong-ro, Kyeong-gi, Pocheon 11160, Korea; eunjinlimsydney@gmail.com; 2Department of Korean Medicine, Institute of Bioscience and Integrative Medicine, Daejeon University, 62 Daehak-ro, Dong-gu, Daejeon 300-716, Korea

**Keywords:** chronic fatigue syndrome, CFS, ME/CFS, prevalence, Korea and Japan, meta-analysis

## Abstract

Background: Myalgic encephalomyelitis/chronic fatigue syndrome (ME/CFS) is a long-term disabling illness accompanied by fatigue unsolved by rest. However, ME/CFS is a poorly understood illness that lacks a universally accepted pathophysiology and treatment. A lack of CFS-related studies have been conducted in Asian countries. This study aimed to estimate and compare the prevalence of ME/CFS in Korea and Japan and conducted a meta-analysis. Methods: We searched PubMed, EMBASE, Cochrane, and KMBASE for population-based prevalence studies of the two countries and synthesized the data according to the Fukuda case definition. Results: Of the eight studies (five in Korea, three in Japan) included, the total prevalence rate of Korean studies was 0.77% (95% CI 0.34–1.76), and 0.76% (95% CI 0.46–1.25) for the Japanese studies. The prevalence rate in females was approximately two-fold higher than males in Korean studies (1.31% female vs. 0.60% male), while the gender difference was less obvious in Japanese studies (0.76% female vs. 0.65% male). Conclusions: Further epidemiology studies on the female ME/CFS prevalence rate between countries may be required.

## 1. Introduction

Fatigue is a common complaint in both the general population and people with disorders [1]. Fatigue generally disappears after rest or treatment; however, uncontrolled chronic fatigue, particularly when lacking a medical explanation, substantially impairs health-related quality of life [2]. Among fatigue-related disorders, chronic fatigue syndrome (CFS) is the most debilitating, resulting in unemployment in half of the affected patients and a seven-fold higher risk of suicide compared to healthy subjects [3]. Until recently, there has been trivialization of CFS, with a debate on the origin of the illness (psychological vs. neurological) [4]. Currently, the Institute of Medicine (IOM) in the U.S. defines CFS as a complex multisystem neurological disorder [5].

The CFS prevalence may vary depending on country, ethnicity, sex, age, and especially diagnostic criteria [6]. To date, the prevalence of CFS has been found to be approximately 0.9%, and there are 1.5 to 2 times more women than men affected by CFS worldwide [7]. Most CFS-related studies have been conducted in the United States and United Kingdom, and the prevalence in Asian populations is uncertain due to the lack of studies. Korea and Japan are located near each other, but their ethnicities differ, and some epidemiologic differences in diseases have been reported between the two countries [8].

In order to expand the demographic knowledge of CFS, we sought to estimate and compare the prevalence of CFS in the Korean and Japanese populations.

## 2. Methods

### 2.1. Data Sources and Keywords

Using a domestic database (KMBASE, https://kmbase.medric.or.kr/ (accessed on 4 January 2021)) and three international databases (PubMed, EMBASE, Cochrane), we searched for publications reporting the prevalence of CFS in Japan and Korea and then performed a meta-analysis. The search term was “[[Chronic fatigue syndrome [MeSH term] AND Prevalence] AND [Japan OR Korea]]” and included studies published between January 1994 and January 2021 due to the Fukuda case definition, which was developed in 1994. All references listed in any included study were also then searched to identify titles matching our study question.

### 2.2. Eligibility and Exclusion Criteria

Papers were initially assessed according to the inclusion criteria. After reading the title and abstract, full articles that met the criteria were thoroughly read and screened by the exclusion criteria. The inclusion criteria were a population-based clinical study, and a prevalence study of ME/CFS in Japan and Korea. For consistency of the data, we included studies used only the Fukuda criteria (also known as CDC-1994 criteria). The exclusion criteria were nonclinical-based studies, randomized controlled studies, and studies focusing on clinical features or biological aspects of ME/CFS (Figure 1).

### 2.3. Review Process

Two authors searched databases and selected the eligible articles according to the above criteria. The prevalence rate by country, the number of participants and patients, and their gender were collected from the selected articles. The final decision in extracting the data was made based upon the consensus of the two authors.

### 2.4. Statistical Analysis

The number of participants and ME/CFS patients from the included articles were organized to compare Korea vs. Japan and male vs. female to estimate the number of populations and the prevalence. A meta-analysis using the random effects model by the Comprehensive Meta Analysis (CMA) program was conducted to estimate heterogeneity of the data.

## 3. Results

### 3.1. Study Characteristics

In total, eight studies, five in Korea [9,10,11,12,13] and three in Japan [14,15,16] were included, which involved a total of 72,669 participants (6319 Korean and 66,350 Japanese). The sex ratios (male vs. female) of participants were very similar in both Korean studies (52% vs. 48%) and Japanese studies (50% vs. 50%), but the average number of participants was much higher in studies of Japanese (22,365 ± 37,369) than Korean (1264 ± 496). The mean age of participants was younger in Korean studies (30.3 ± 11.9 years) than Japanese (46.7 ± 10.4 years) (Table 1).

### 3.2. CFS Prevalence

From the results of the meta-analysis, the overall prevalence rate of CFS in the Korean studies was 0.77% (95% CI 0.34–1.76), being slightly higher than the 0.76% reported in the Japanese studies (95% CI 0.46–1.25) (Figure 2). As expected, there was an approximately two-fold female predominance (1.31% female vs. 0.60% male) in Korean studies, while this sex difference was less obvious in Japanese studies, with 0.76% female and 0.65% male (Figure 3 and Figure 4).

## 4. Discussion

As we expected, the prevalence rate in Korea (0.77%) and Japan (0.76%) of CFS was similar to those already reported worldwide (approximately 1%) [17]. We previously reported that CFS prevalence could be strongly affected by the diagnostic method or case definition used; the difference between using the Fukuda (CDC-1994) and Holmes (CDC-1988) criteria yielded rates of 0.89% vs. 0.17%, respectively [6]. In the present analysis, all the included eight studies used the Fukuda criteria. The Fukuda criteria relies on the presence of four or more of the eight main symptoms [18], which are the most commonly used criteria for the diagnosis of CFS in clinics including clinical trials [19].

All eight studies used interviews as the diagnostic method, and four studies additionally conducted or included the results of medical tests according to CDC-1994 guideline [18], three of the five Korean studies [9,10,11] and one of the three Japanese studies [16]. This means that Korean studies applied stricter criteria for the diagnosis of CFS than Japanese studies, which can be considered as leading to a reduction in the CFS prevalence rate of Korean data. However, in this study, we found the prevalence rates in Korea (0.77%, 95% CI 0.34–1.76) and Japan (0.76%, 95% CI 0.46–1.25) were very similar. This result is somewhat different from our previous meta-analysis for global CFS prevalence; we found the slight difference in the prevalence rate depends on the diagnostic methods, for example: diagnosed by primarily interview (1.14%) vs. interview conducted with medical tests (0.95%) [6]. This might tell the fact that inclusion of medical tests for CFS diagnosis may not have much of an impact on the overall prevalence rate. Lim et al.’s study included a total of 28 studies (interviews, 19 studies, vs. interviews with medical tests, nine studies) [6], whereas only eight (four vs. four, respectively) were included in this study (Table 1). Additionally, the prevalence rate among the studies using the same diagnostic method, that is with medical tests of the primary care participants, showed widely varied prevalence rates (ex, 0.61% Kim CH 2005 vs. 2.03% Ji JD 2000) between the studies [10,11]. In addition to the diagnostic method, race as a factor is another important aspect of CFS prevalence. This study shows that the meta-analysis results of each study in Korea and Japan were widely varied, ranging from 0.06% to 1.46% for the subjects diagnosed by interview (four studies), and 0.61% to 1.89% (four studies) for interviews with medical tests. We compared these results to the studies of Western countries (mainly the UK and USA), using CDC-1994 and an adult general population recruited from both primary care and community settings. Those results ranged from 0.23% to 2.74% (five studies) and 0.42% to 2.62% (six studies), which indicates a somewhat higher prevalence than in Korea and Japan [6]. Being the female gender is a well-known predisposing factor of CFS [20], and an approximate two-fold female predominance of CFS was observed in Korean studies, while this female predominance was very low in Japanese studies (0.76% female and 0.65% male). We could anticipate that the large difference in CFS prevalence in females between Korea and Japan (1.31% vs. 0.76%) might be due to genetic and/or social factors. Comparative studies have reported slightly higher prevalence rates of four noncommunicable diseases in Korea than Japan, including hypertension (17.6% in Korea vs. 15.2% in Japan), diabetes (5.7% vs. 4.8%), hyperlipidemia (9.2% vs. 6.9%), and angina pectoris, and their general risk factors (1.7% vs. 1.5%). However, the sex-related risk was different according to the disease; for example, the male-to-female odds ratio was 0.88 in Korea vs. 0.64 in Japan for hypertension but 1.05 vs. 1.39 for diabetes [21].

The present study has some limitations: the original epidemiologic studies included were outdated (conducted 10 to 25 years ago) using the CDC-1994 case definition, and the average sample size, especially in Korean studies, was relatively small (1264 ± 496). One Japanese study was performed on a large scale (65,500 participants) [14], but the other two studies (1430 and 147 participants) were small. The results of a large gender difference are limited to Korea and Japan in this study, which should not be generalized to world data.

Despite those limitations, we systematically estimated the overall prevalence rate of CFS in Korea and Japan and identified differences. This information would be helpful for epidemiological research on CFS in the future. Further studies may be required on the rationales for the genetic and social factors that cause the gender difference in prevalence rate of CFS between Korea and Japan. Moreover, investigation of the prevalence rate using updated case definitions may also be needed.

## Figures and Tables

**Figure 1 jcm-10-03204-f001:**
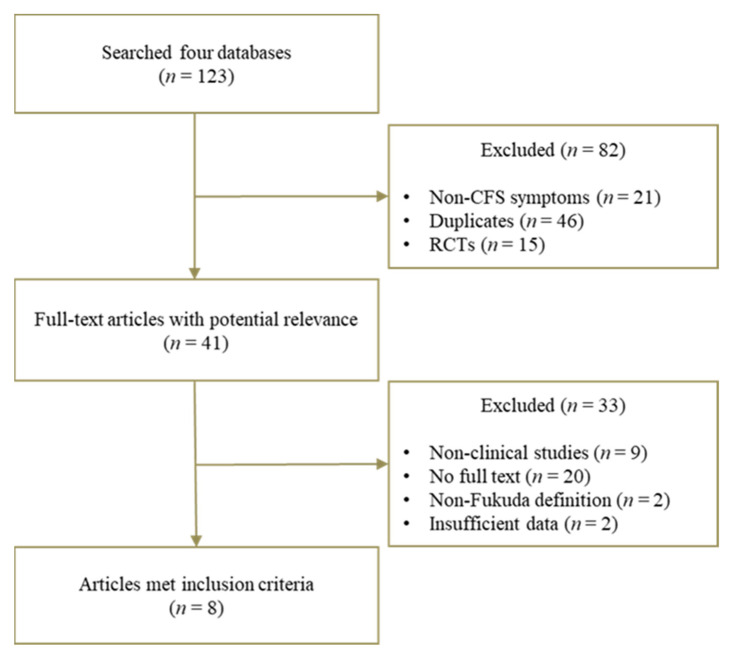
Flow chart for selection of articles.

**Figure 2 jcm-10-03204-f002:**
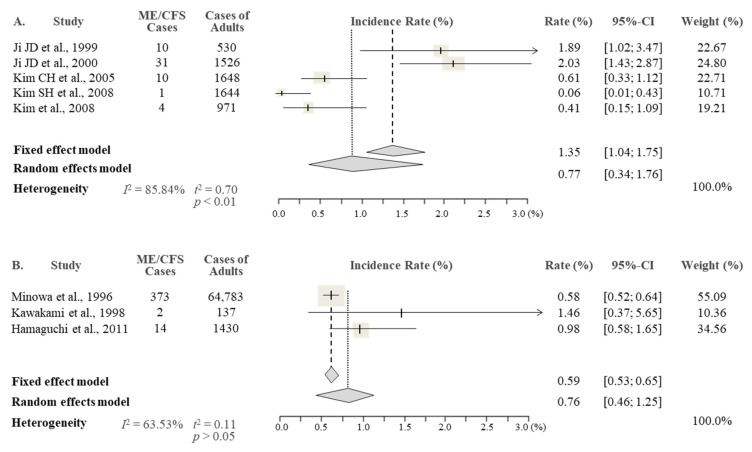
Meta-analysis of the ME/CFS total prevalence in Korea (**A**) and Japan (**B**).

**Figure 3 jcm-10-03204-f003:**
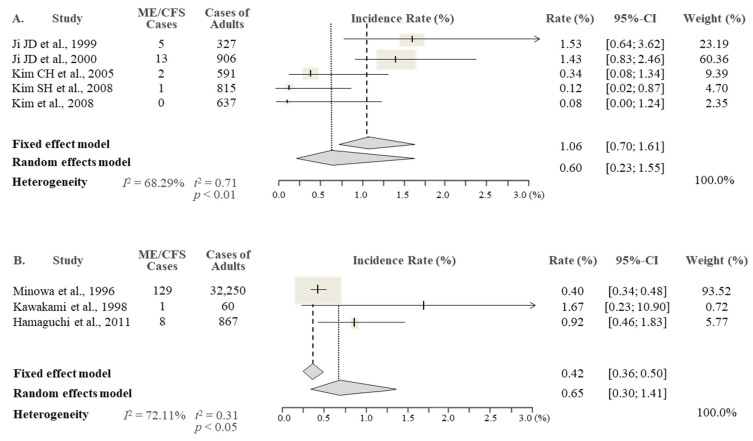
Meta-analysis of the ME/CFS prevalence in males in Korea (**A**) and Japan (**B**).

**Figure 4 jcm-10-03204-f004:**
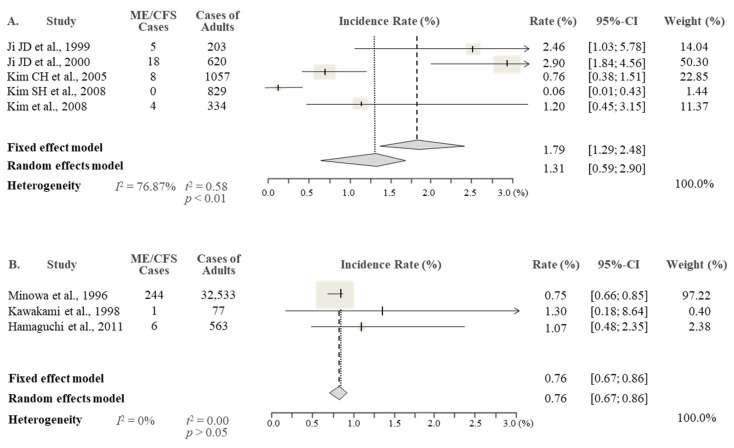
Meta-analysis of the ME/CFS prevalence in females in Korea (**A**) and Japan (**B**).

**Table 1 jcm-10-03204-t001:** Characteristics of the studies reporting the prevalence of ME/CFS in Korea and Japan.

Group	Korea	Japan	Total
Number of studies included (%)	5 (63)	3 (37)	8 (100)
Community	2 (25)	2 (25)	4 (50)
Primary care	3 (38)	1 (12)	4 (50)
Total number of participants	6319	66,350	72,669
(Average ± SD)	(1264 ± 496)	(22,365 ± 37,369)	(9173 ± 22,766)
Number of participants Male/Female	3276/3043	33,177/33,173	36,453/36,216
(male: female Ratio %)	(52:48)	(50:50)	(50:50)
Mean age of participants ^±^	30.3 ± 11.9	46.7 ± 10.4	38.5 ± 14.9
Year of publication (N. of participants)			
1994–2000	2 (2056)	2 (64,920)	4 (66,976)
2001–2010	3 (4263)		3 (4263)
2011–2018		1 (1430)	1 (1430)
Diagnostic method			
Interview (medical test −)	2 (2615)	2 (64,920)	4 (67,535)
Interview (medical test +)	3 (3704)	1 (1430)	4 (5134)

^±^ Estimated from either the reported mean age for each gender or the mean age of both sexes. The age of children and adolescents was excluded.

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
