# Peer review of "Prevalence of Chronic Fatigue Syndrome (CFS) in Korea and Japan: A Meta-Analysis"

_jcm, 2021, doi:10.3390/jcm10153204_

Round 1

Reviewer 1 Report

In overall, the manuscript is relatively short and concise (in form of a brief communication). Yet, it contains very important information regarding prevalence of ME/CFS in areas of worlds, that it was not previously determined in such comprehensive way. The whole manuscript it really easy to follow. I have found just two major issues and a few minor:

Two major issues:

  1. Editing of manuscript should be improved, especially in the results section.
  2. Discussion should be extended a little. I would extend the second paragraph of discussion citing other researches to compare the prevalence of ME/CFS in areas that it has been determined (for instance: Europe (UK, Poland) Australia, etc.). In addition, what is repeated over and over in literature on ME/CFS is the fact of alleged higher prevalence of ME/CFS in white/Caucasian population. Would You like to add a sentence on this: how Your results could be compared with previous studies that were including race as a factor? Maybe it is worth to add at the end of manuscript in terms of future studies that should examine it further?

Minor issues:

Abstract:

Lines 15-16 „Most CFS-related 15 studies have been conducted in the Western countries, and only the lack of Asian studies have been 16 published” I do agree, but please reword this sentence.

“The large difference of fe-24 male prevalence rate between Korea and Japan may be associated with genetic and/or social factors.” Very interesting conclusion but please be more specific. I suppose that Your data do not support such conclusion, so maybe it would be better to omit that one from abstract, leaving those conclusions that are related to Your data? The same sentence appears in discussion, but I would leave it there because in the discussion You explained the background for this.

Main text:

„Figure 1. Flow chart for selection of articles” in my humble opinion would be more clear if all horizontal arrows would be only pointing to right

Line 127 “Our results from Korea and Japan were similar to the overall global findings re-127 garding the prevalence of CFS, with the exception of the relatively small sex difference in 128 Japan.” This is a very important conclusion but should be moved to the discussion section, where other studied should be cited for comparison

Discussion “previous meta-analysis for global CFS prevalence, we found the slight difference of the 155 prevalence rate, depending on the diagnostic methods, namely interview without (1.14%) 156 vs. with (0.95%) medical tests [6]” could You elaborate on that a little? What does it mean „interview with medical tests” exactly?

Author Response

Reviewer 1.

1-1. Editing of manuscript should be improved, especially in the results section.

→ Thank reviewer for the comments. I have made amendment for the results section. please refer to page 3-5.

1-2. Discussion should be extended a little. I would extend the second paragraph of discussion citing other researches to compare the prevalence of ME/CFS in areas that it has been determined (for instance: Europe (UK, Poland) Australia, etc.). In addition, what is repeated over and over in literature on ME/CFS is the fact of alleged higher prevalence of ME/CFS in white/Caucasian population. Would You like to add a sentence on this: how Your results could be compared with previous studies that were including race as a factor? Maybe it is worth to add at the end of manuscript in terms of future studies that should examine it further?

→ Thank reviewer for the crucial comments. As reviewer recommended, I have extended the discussion part by adding “In addition to the diagnostic method, race as a factor is another important aspect of CFS prevalence. This study shows that the meta-analysis results of each study in Korea and Japan, was widely varied ranged from 0.06% to 1.46% for the subjects diagnosed by interview (4 studies), and 0.61% to 1.89% (4 studies) for interview with medical tests. We compared these results to the studies of western countries (mainly UK and USA), used CDC-1994, adult general population recruited from both primary care and community settings. Those results were ranged 0.23% to 2.74% (5 studies) and 0.42% to 2.62% (6 studies), which indicated somewhat higher prevalence than Korea and Japan.”, on page 6.

1-3. Abstract: Lines 15-16 “Most CFS-related studies have been conducted in the Western countries, and only the lack of Asian studies have been published” I do agree, but please reword this sentence.

→ Thank reviewer for your detail correction. We have amended the sentence to “Lack of CFS-related studies have been conducted in Asian countries.”

1-4. Abstract: “The large difference of fe-24 male prevalence rate between Korea and Japan may be associated with genetic and/or social factors.” Very interesting conclusion but please be more specific. I suppose that Your data do not support such conclusion, so maybe it would be better to omit that one from abstract, leaving those conclusions that are related to Your data? The same sentence appears in discussion, but I would leave it there because in the discussion You explained the background for this.

→ Thank reviewer for the professional and helpful comments and suggestions. We deleted the sentence in the abstract, because we do agree that our data is insufficient to conclude ‘genetic and/or social factors’, and the following sentence “Further epidemiology studies on the female ME/CFS prevalence rate between countries may be required” would also highlight the prominence of female.

1-5. Main text: Figure 1. Flow chart for selection of articles” in my humble opinion would be more clear if all horizontal arrows would be only pointing to right

→ Thank reviewer for your kind suggestion. We have amended the flow chart as you recommended.

1-6. Main text: Line 127 “Our results from Korea and Japan were similar to the overall global findings re-127 garding the prevalence of CFS, with the exception of the relatively small sex difference in 128 Japan.” This is a very important conclusion but should be moved to the discussion section, where other studied should be cited for comparison

→ Thank reviewer for your thorough review. We deleted the sentence in the Results, as it is stated in the Discussion section on page 5 and 6.

1-7. Main text: Discussion “previous meta-analysis for global CFS prevalence, we found the slight difference of the 155 prevalence rate, depending on the diagnostic methods, namely interview without (1.14%) 156 vs. with (0.95%) medical tests [6]” could You elaborate on that a little? What does it mean „interview with medical tests” exactly?

→ Thank reviewer for your comment. We have amended the description, as “We found the slight difference of the prevalence rate, depending on the diagnostic methods, for example: diagnosed primarily by interview (1.14%) vs. interview conducted with medical tests (0.95%).” on page 6.

Reviewer 2 Report

This is a straightforward meta-analysis of some reports of CFS prevalence in Korea and Japan. It seems to me the main surprise is the lack of a greater gender difference in Japan.  This will be a useful addition to the literature on prevalence of pre-2020 ME/CFS, as some groups are now claiming long COVID will be diagnosed as ME/CFS.  I would suggest to the authors that they put their work in context in the discussion by discussing and citing two relevant papers that are quite relevant: https://pubmed.ncbi.nlm.nih.gov/30671425/ and https://pubmed.ncbi.nlm.nih.gov/32466160/.

This is a suggestion, not essential, but may increase the reach of their article.

Author Response

Reviewer 2

2-1. This is a straightforward meta-analysis of some reports of CFS prevalence in Korea and Japan. It seems to me the main surprise is the lack of a greater gender difference in Japan.  This will be a useful addition to the literature on prevalence of pre-2020 ME/CFS, as some groups are now claiming long COVID will be diagnosed as ME/CFS.  I would suggest to the authors that they put their work in context in the discussion by discussing and citing two relevant papers that are quite relevant: https://pubmed.ncbi.nlm.nih.gov/30671425/ and https://pubmed.ncbi.nlm.nih.gov/32466160/.

This is a suggestion, not essential, but may increase the reach of their article.

→  Thank reviewer for your kind suggestion. This study is describing the results of only Korea and Japan. Thus, the large gender difference is limited to the two countries. Your kind suggestion to discuss the two relevant studies are referred to our previous study of prevalence of CFS in worldwide, as those two studies are discussing gender difference for world prevalence rate. Additionally, we have added this in the limitation “The results of large gender difference are limited to Korea and Japan in this study, which should not be generalized to world data.” on page 6.

Reviewer 3 Report

Many thanks for the update with regard to the peer review of the manuscript.  I/we welcome this paper, as it is one of the few manuscripts which deals with ME/CFS in Japan and Korea.  Currently, there is a dearth of literature describing the epidemiology of ME/CFS in these countries.  The meta-analysis performed seems to be appropriate.   There are, however, several nuances with regard to ME/CFS that seem to have been missed, and which I think would be important to address:   Fukuda et al., defines a now, outdated research case definition which was never intended for clinical practice.  Criticism has been raised that Fukuda et al., is overly broad in its definition and may include patients who would not currently be thought of as having ME/CFS.   During the time span covered by the meta-analysis of this manuscript, newer case definitions for ME/CFS have evolved, further antiquating the use of Fukuda et al.   The authors should address these issues, at least acknowledge their existence, and since they mention the Institute of Medicine Report of 2015, discuss the concerns of the ME/CFS case definition as articulated in that report.   I can well understand why the authors chose to use the Fukuda et al. case definition for their meta-analysis, but they do need to put that case definition in context and discuss the limitations of having done so to our current understanding of ME/CFS.   Many thanks for the opportunity to provide some input.

Author Response

Reviewer 3

3-1. Many thanks for the update with regard to the peer review of the manuscript.  I/we welcome this paper, as it is one of the few manuscripts which deals with ME/CFS in Japan and Korea.  Currently, there is a dearth of literature describing the epidemiology of ME/CFS in these countries.  The meta-analysis performed seems to be appropriate.   There are, however, several nuances with regard to ME/CFS that seem to have been missed, and which I think would be important to address:   Fukuda et al., defines a now, outdated research case definition which was never intended for clinical practice.  Criticism has been raised that Fukuda et al., is overly broad in its definition and may include patients who would not currently be thought of as having ME/CFS.   During the time span covered by the meta-analysis of this manuscript, newer case definitions for ME/CFS have evolved, further antiquating the use of Fukuda et al. The authors should address these issues, at least acknowledge their existence, and since they mention the Institute of Medicine Report of 2015, discuss the concerns of the ME/CFS case definition as articulated in that report.   I can well understand why the authors chose to use the Fukuda et al. case definition for their meta-analysis, but they do need to put that case definition in context and discuss the limitations of having done so to our current understanding of ME/CFS.   Many thanks for the opportunity to provide some input.

→ Thank reviewer for your comment. We have amended the limitation part in Discussion, as “The present study has some limitations: the original epidemiologic studies included were outdated (conducted 10 to 25 years ago) using CDC-1994 case definition, ~~~~ “ on page 6. Also, we have added “Moreover, investigation of prevalence rate using updated case definitions may also be needed.” for further study on page 7.

Yours sincerely,

Eun Jin Lim

This manuscript is a resubmission of an earlier submission. The following is a list of the peer review reports and author responses from that submission.